# Reward-Predictive Clustering

## Abstract

Recent advances in reinforcement-learning research have demonstrated impressive results in building algorithms that can out-perform humans in complex tasks. Nevertheless, creating reinforcement-learning systems that can build abstractions of their experience to accelerate learning in new contexts still remains an active area of research. Previous work showed that reward-predictive state abstractions fulfill this goal, but have only be applied to tabular settings. Here, we provide a clustering algorithm that enables the application of such state abstractions to deep learning settings, providing compressed representations of an agent's inputs that preserve the ability to predict sequences of reward. A convergence theorem and simulations show that the resulting reward-predictive deep network maximally compresses the agent's inputs, significantly speeding up learning in high dimensional visual control tasks. Furthermore, we present different generalization experiments and analyze under which conditions a pre-trained reward-predictive representation network can be re-used without re-training to accelerate learning—a form of systematic out-of-distribution transfer.

## 1 Introduction

Recent advances in reinforcement learning (RL) (Sutton & Barto, 2018) have demonstrated impressive results, outperforming humans on a range of different tasks (Silver et al., 2016; 2017b; Mnih et al., 2013). Despite these advances, the problem of building systems that can re-use knowledge to accelerate learning—a characteristic of human intelligence—still remains elusive. By incorporating previously learned knowledge into the process of finding a solution for a novel task, intelligent systems can speed up learning and make fewer mistakes. Therefore, efficient knowledge re-use is a central, yet under-developed, topic in RL research.

We approach this question through the lens of representation learning. Here, an RL agent constructs a representation function to compress its high-dimensional observations into a lower-dimensional latent space. This representation function allows the system to simplify complex inputs while preserving all information relevant for decision-making. By abstracting away irrelevant aspects of task, an RL agent can efficiently generalize learned values across distinct observations, leading to faster and more data-efficient learning (Abel et al., 2018; Franklin & Frank, 2018; Momennejad et al., 2017). Nevertheless, a representation function can become specialized to a specific task, and the information that needs to be retained often differs from task to task. In this context, the question of how to compute an efficient and *re-usable* representation emerges.

In this article, we introduce a clustering algorithm that computes a reward-predictive representation (Lehnert et al., 2020; Lehnert & Littman, 2020) from a fixed data set of interactions—a setting commonly known as *offline RL* (Levine et al., 2020). A reward-predictive representation is a type of function that compresses high-dimensional inputs into lower-dimensional latent states. These latent states are constructed such that they can be used to predict future rewards without having to refer to the original high dimensional input. To compute such a representation, the clustering algorithm processes an interaction data set that is sampled from a single *training task*. First, every state observation is assigned to the same latent state index. Then, this single state cluster is iteratively refined by introducing additional latent state indices and re-assigning some state observations to them. At the end, the assignment between state observations and latent state cluster indices can be used to train a representation network that classifies high-dimensional states into one of the computed latent state cluster. Later on, the output of this representation network can be used to predict future reward outcomes without referring to the original high-dimensional state. Therefore, the

resulting representation network is a reward-predictive representation. The presented clustering algorithm is generic: Besides constraining the agent to decide between a finite number of actions, no assumptions about rewards or state transitions are made. We demonstrate that these reward-predictive representation networks can be used to accelerate learning in *test tasks* that differ in both transition and reward functions from those used in the training task. The algorithm demonstrates a form of out-of-distribution generalization because the test tasks require learning a task solution that is novel to the RL agent and does not follow the training data's distribution. The simulation experiments reported below demonstrate that reward-predictive representation networks comprise a form of abstract knowledge re-use, accelerating learning to new tasks. To unpack how reward-predictive representation networks can be learned and transferred, we first illustrate the clustering algorithm using different examples and prove a convergence theorem. Lastly, we present transfer experiments illuminating the question of when the learned representation networks generalize to test tasks that are distinct from the training task in a number of different properties.

## 2 Reward-predictive representations

Mathematically, a reward-predictive representation is a function $\boldsymbol{\phi}$ that maps an RL agent's observations to a vector encoding the compressed latent state. Figure 1 illustrates a reward-predictive representation with an example. In the Column World task (Figure 1(a)), an RL agent navigates through a grid and receives a reward every time a green cell (right column) is entered. Formally, this task is modelled as a Markov Decision Process (MDP) $M = \langle \mathcal{S}, \mathcal{A}, p, r \rangle$, where the set of observations or *states* is denoted with $\mathcal{S}$ and the finite set of possible *actions* is denoted with $\mathcal{A}$. The transitions between adjacent grid cells are modelled with a transition function $p(s, a, s')$ specifying the probability or density function of transitioning from state $s$ to state $s'$ after selecting action $a$. Rewards are specified by the reward function $r(s, a, s')$ for every possible transition.

To solve this task optimally, the RL agent needs to know which column it is in and can abstract away the row information from each grid cell. (For this example we assume that the abstraction is known; the clustering algorithm below will show how it can be constructed from data). Figure 1(b) illustrates this abstraction as a state colouring: By assigning each column a distinct colour, the $4 \times 4$ grid can be abstracted into a $4 \times 1$ grid. A representation function then maps every state in the state space $\mathcal{S}$ to a latent state vector (a colour). Consequently, a trajectory (illustrated by the black squares and arrows in Figure 1(b)) is then mapped to a trajectory in the abstracted task. The RL agent can then associate colours with decisions or reward predictions instead of directly operating on the higher-dimensional $4 \times 4$ grid.

This colouring is a reward-predictive representation, because for any arbitrary start state and action sequence it is possible to predict the resulting reward sequence using only the abstracted task. Formally, this can be described by finding a function $f$ that maps a start latent state and action sequence to the expected reward sequence:

$$f(\boldsymbol{\phi}(s), a_1, ..., a_n) = \mathbb{E}_p\left[(r_1, ..., r_n)|s, a_1, ..., a_n\right]. \tag{1}$$

The right-hand side of Equation (1) evaluates to the expected reward sequence observed when following the action sequence $a_1, ..., a_n$ starting at state $s$ in the original task. The left-hand side of Equation (1) predicts this reward sequence using the action sequence $a_1, ..., a_n$ and only the latent state $\boldsymbol{\phi}(s)$—the function $f$ does not have access to the original state $s$. This restricted access to latent states constrains the representation function $\boldsymbol{\phi}$ to be reward-predictive in a specific task: Given the representation's output $\boldsymbol{\phi}(s)$ and not the full state $s$, it is possible to predict an expected reward sequence for any arbitrary action sequence using a latent model $f$. Furthermore, once an agent has learned how to predict reward-sequences for one state, it can re-use the resulting function $f$ to immediately generalize predictions to other states that map to the same latent state, resulting in faster learning. Of course, a reward-predictive representation always encodes some abstract information about the task in which it was learned; if this information is not relevant in a subsequent task, an RL agent would have to access the original high-dimensional state and learn a new representation. We will explore the performance benefits of re-using reward-predictive representations empirically in Section 4. The colouring in Figure 1(b) satisfies the condition in Equation (1): By associating green with a reward of one and all other colours with a reward of zero, one can use only a start colour and action sequence to predict

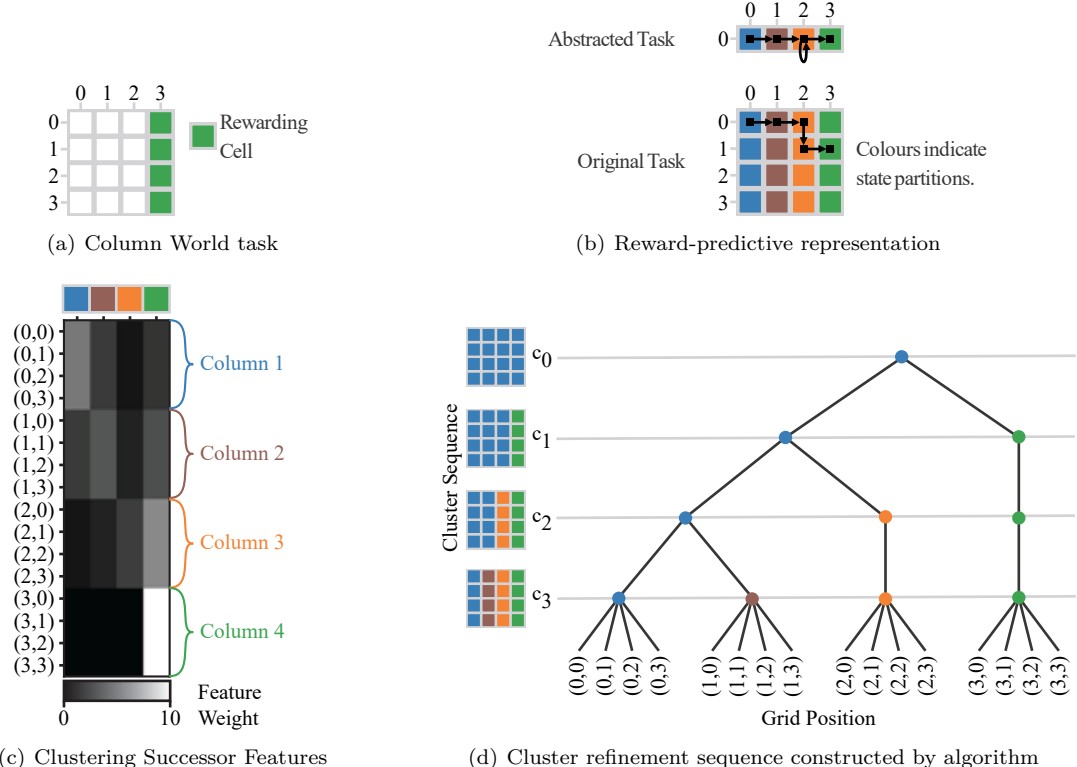

(a) Column World task

(b) Reward-predictive representation

(c) Clustering Successor Features

(d) Cluster refinement sequence constructed by algorithm

Figure 1: Reward-predictive clustering in the Column World task. (a): In the Column World task the agent can transition between adjacent grid cells by selecting from one of four available actions: move up, down, left, or right. A reward of one is given if a green cell is given, otherwise rewards are zero. All transitions are deterministic in this task. (b): By colouring every column in a distinct colour, every state of the same column is assigned the same latent state resulting in a $4 \times 1$ abstracted grid world task. In this example, an agent only needs to retain which column it is in to predict future rewards and can therefore only use the abstracted task to predict reward sequences for every possible trajectory. (c): Matrix plot of all SF vectors $\boldsymbol{\psi}^{\pi}(s, a)$ for the move "move right" action an a policy $\pi$ that selects actions uniformly at random. Every row corresponds to the four-dimensional vector for each grid position, as indicated by the y-axis labels. For this calculation, the colour of a state $s$ is encoded as a colour index $c(s)$ that ranges from one through four and the state-representation vector is a one-hot bit vector $\boldsymbol{e}_{c(s)}$ where the entry $c(s)$ is set to one and all other entries are set to zero. (d): Colour function sequence $c_0, c_1, c_2, c_3$ generated by the reward-predictive clustering algorithm. Initially, all states are merged into a single partition and this partitioning is refined until a reward-predictive representation is obtained. The first clustering $c_1$ is obtained by associating states with equal one-step rewards with the same colour (latent state vector). Then, the SF matrix shown in (c) is computed for a state representation that associates state with the blue-green colouring as specified by $c_1$. The row space of this SF matrix is then clustered again leading to the clustering $c_2$. Subsequently, the SF matrix is computed again for the blue-orange-green colouring and the clustering procedure is repeated. This method iteratively refines a partitioning of the state space until a reward-predictive representation is obtained.

a reward sequence and this example can be repeated for every possible start state and action sequence of any length.

## 2.1 Improving learning efficiency with successor representations

To improve an RL agent's ability to generalize its predictions across states, the Successor Representation (SR) was introduced by Dayan (1993). Instead of explicitly planning a series of transitions, the SR summarizes the frequencies with which an agent visits different future states as it behaves optimally and maximizes rewards. Because the SR models state visitation frequencies, this representation implicitly encodes the task's transition function and optimal policy. Consequently, the SR provides an intermediate between model-based RL, which focuses on learning a full model of a task's transition and reward functions, and model-free RL, which focuses on learning a policy to maximize rewards (Momennejad et al., 2017; Russek et al., 2017). Barreto et al. (2017) showed that the SR can be generalized to Successor Features (SFs), which compress the high dimensional state space into a lower dimensional one that can still be used to predict future state occupancies. They demonstrated how SFs can be re-used across tasks with different reward functions to speed up learning. Indeed, SFs—like the SR—only reflect the task's transition function and optimal policy but are invariant to any specifics of the reward function itself. Because of this invariance, SFs provide an initialization allowing an agent to adapt a previously learned policy to tasks with different reward functions, leading to faster learning in a life-long learning setting (Barreto et al., 2018; 2020; Lehnert et al., 2017; Nemecek & Parr, 2021).

However, such transfer requires the optimal policy in the new task to be similar to that of the previous tasks (Lehnert & Littman, 2020; Lehnert et al., 2020). For example, even if only the reward function changes, but the agent had not typically visited states near the new reward location in the old task, the SR/SF is no longer useful and must be relearned from scratch (Lehnert et al., 2017). To further improve the invariance properties of SFs, Lehnert & Littman (2020) presented a model that makes use of SFs solely for establishing which states are equivalent to each other for the sake of predicting future reward sequences, resulting in a reward-predictive representation. Because reward-predictive representations only model state equivalences, removing the details of exactly how (i.e., they are invariant to the specifics of transitions, rewards, and the optimal policy), they provide a mechanism for a more abstract form of knowledge transfer across tasks with different transition and reward functions (Lehnert & Littman, 2020; Lehnert et al., 2020). Formally, SFs are defined as the expected discounted sum of future latent state vectors and

$$\boldsymbol{\psi}^{\pi}(s, a) = \mathbb{E}_{a,\pi} \left[ \sum_{t=1}^{\infty} \gamma^{t-1} \boldsymbol{\phi}(s_t) \middle| s_1 = s \right], \tag{2}$$

where the expectation in Equation (2) is calculated over all infinite length trajectories that start in state $s$ with action $a$ and then follow the policy $\pi$. The connection between SFs and reward-predictive representations is illustrated in Figure 1(c). Every row in the matrix plot in Figure 1(c) shows the SF vector $\boldsymbol{\psi}^{\pi}(s, a)$ for each of the 16 states of the Column World task. One can observe that states belonging to the same column have equal SFs. Lehnert & Littman (2020) prove that states that are mapped to the same reward-predictive latent state (and have therefore equal colour) also have equal SFs. In other words, there exists a bijection between two states that are equivalent in terms of their SF vectors and two states belonging to the same reward-predictive latent state.

As such, previous work (Lehnert et al., 2020; Lehnert & Littman, 2020; 2018) computes a reward-predictive representation for finite MDPs by optimizing a linear model using a least-squares loss objective. This loss objective requires the representation function $\boldsymbol{\phi}$ to be linear in the SFs and reward function. Furthermore, it scores the accuracy of SF predictions using a mean-squared error. These two properties make it difficult to directly use this loss objective for complex control tasks, because SFs may become very high dimensional and it may be difficult to predict individual SF vectors with near perfect accuracy while also obtaining a representation function that is linear in these predictions. This issue is further exacerbated by the fact that in practice better results are often obtained by training deep neural networks as classifiers rather than regressors of complex or sparse functions. Additionally, in this prior approach the degree of compression was specified using a hyper-parameter by a human expert. Here, we present a clustering algorithm that remedies

136 these three limitations by designing a cluster-refinement algorithm instead of optimizing a parameterized
137 model with end-to-end gradient descent. Specifically, the refinement algorithm implicitly solves the loss
138 objective introduced by Lehnert & Littman (2020) in a manner similar to temporal-difference learning
139 or value iteration. Initially, the algorithm starts with a parsimonious representation in which all states
140 are merged into a single latent state cluster and then the state representation is iteratively improved by
141 minimizing a temporal difference error defined for SF vectors. This is similar to value iteration or temporal-
142 difference learning, whereby values are assumed to be all zero initially but then adjusted iteratively, but here
143 we apply this idea to refining a state representation (Figure 1(d)). Through this approach, we avoid having
144 to optimize a model with a linearity constraint as well as using a least-squared error objective to train a
145 neural network. Instead, the clustering algorithm only trains a sequence of state classifiers to compute a
146 reward-predictive representation. Furthermore, the degree of compression—the correct number of reward-
147 predictive latent states—is automatically discovered. This is accomplished by starting with a parsimonious
148 representation in which all states are merged into a single latent state cluster and iteratively improving the
149 state representation until a reward-predictive representation is obtained without adding any additional latent
150 states in the process. In the following section, Section 3, we will formally outline how this algorithm computes
151 a reward-predictive state representation and discuss a convergence proof. Subsequently, we demonstrate
152 how the clustering algorithm can be combined with deep learning methods to compute a reward-predictive
153 representation for visual control tasks (Section 4). Here, we analyze how approximation errors contort the
154 resulting state representation. Lastly, we demonstrate how reward-predictive representation networks can be
155 used to accelerate learning in tasks where an agent encounters both novel state observations and transition
156 and reward functions.

## 3 Iterative partition refinement

158 The reward-predictive clustering algorithm receives a fixed trajectory data set

$$\mathcal{D} = \{(s_{i,0}, a_{i,0}, r_{i,0}, s_{i,1}, a_{i,1}, ..., s_{i,L_i})\}_{i=1}^{D} \tag{3}$$

159 as input. Each data point in $\mathcal{D}$ describes a trajectory of length $L_i$. While we assume that this data set $\mathcal{D}$
160 is fixed, we do not make any assumptions about the action-selection strategy used to generate this data set.
161 The clustering algorithm then generates a cluster sequence $c_0, c_1, c_2, ...$ that associates every observed state
162 $s_{i,t}$ in $\mathcal{D}$ with a cluster index. This cluster sequence is generated with an initial reward-refinement step and
163 subsequent SF refinement steps until two consecutive clustering are equal. These steps are described next.

### 3.1 Reward refinement

165 To cluster states by their one-step reward values, a function $f_r$ is learned to predict one-step rewards. This
166 function is obtained through Empirical Risk Minimization (ERM) (Vapnik, 1992) by solving the optimization
167 problem

$$f_r = \arg\min_f \sum_{(s,a,r,s') \in \mathcal{D}} |f(s,a) - r|, \tag{4}$$

168 where the summation ranges over all transitions between states in the trajectory data set $\mathcal{D}$. This optimiza-
169 tion problem could be implemented by training a deep neural network using any variation of the backprop
170 algorithm (Goodfellow et al., 2016). Because rewards are typically sparse in an RL task and because deep
171 neural networks often perform better as classifiers rather than regressors, we found it simpler to first bin the
172 reward values observed in the transition data set $\mathcal{D}$ and train a classifier network that outputs a probability
173 vector over the different reward bins. Instead of using the absolute value loss objective stated in Equation (4),
174 this network is trained using a cross-entropy loss function (Goodfellow et al., 2016). Algorithm 1 outlines
175 how this change is implemented. The resulting function $f_r$ is then used to cluster all observed states by
176 one-step rewards, leading to a cluster assignment such that, for two arbitrary state observations $s$ and $\tilde{s}$,

$$c_1(s) = c_1(\tilde{s}) \implies \sum_{a \in \mathcal{A}} |f_r(s,a) - f_r(\tilde{s},a)| \leq \varepsilon_r. \tag{5}$$

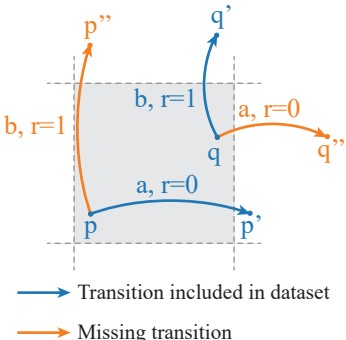

Figure 2: Function approximation is needed to generalize one-step reward predictions and SF predictions for state-action combinations not observed in the transition data set. In this example, the state space consists of points in $\mathbb{R}^2$ and the action space consists of actions $a$ and $b$. We assume that a maximally compressed reward-predictive representation merges all points in the grey square into one latent state. Selecting the action $a$ from within the grey square results in a transition to the right and generates a reward of 0. Selecting the action $b$ from within the grey square results in a transition to the top and generates a reward of 1. If the dataset only contains the two transitions indicated by the blue arrows and the transitions indicated by the orange arrows are missing, then function approximation is used to predict one-step reward predictions and SF for the missing state and action combinations $(p, b)$ and $(q, a)$. These function approximators need to be constrained such that they output the same one-step rewards and SF vectors for points that fall within the shaded square.

Figure 2 illustrates why function approximation is needed to compute the one-step reward clustering in line (5). In this example, states are described as positions in $\mathbb{R}^2$ and all points lying in the shaded area belong to the same partition and latent state. Specifically, selecting action $a$ from within the grey square results in a transition to the right and a reward of zero, while selecting action $b$ results in a transition to the top and a reward of one. We assume that the transition data set only contains the two transitions indicated by the blue arrows. In this case, we have $r(p, a) = 0$ and $r(q, b) = 1$, because $(p, a)$ and $(q, a)$ are state-action combinations contained in the transition data set and a rewards of zero and one were given, respectively. To estimate one-step rewards for the missing state-action combinations $(p, b)$ and $(q, a)$, we solve the function approximation problem in line (4) and then use the learned function $f_r$ to predict one-step reward values for the missing state-action combinations $(p, b)$ and $(q, a)$. For this reward-refinement step to accurately cluster states by one-step rewards, the optimization problem in line (4) needs to be constrained, for example by picking an appropriate neural network architecture, such that the resulting function $f_r$ generalizes the same prediction across the shaded area in Figure 2.

## 3.2 Successor feature refinement

After reward refinement, the state partitions are further refined by first computing the SFs, as defined in Equation (2), for a state representation that maps individual state observations to a one-hot encoding of the existing partitions. Specifically, for a clustering $c_i$ the state representation

$$\boldsymbol{\phi}_i : s \mapsto \boldsymbol{e}_{c_i(s)} \tag{6}$$

is used, where $\boldsymbol{e}_{c_i(s)}$ is a one-hot vector with entry $c_i(s)$ set to one. The individual SF vectors $\boldsymbol{\psi}_i^\pi(s, a)$ can be approximated by first computing a Linear Successor Feature Model (LSFM) (Lehnert & Littman, 2020). The computation results in obtaining a square matrix $\boldsymbol{F}$ and

$$\boldsymbol{\psi}_i^\pi(s, a) \approx \boldsymbol{e}_{c_i(s)} + \gamma \boldsymbol{F} \mathbb{E}_p \left[ \boldsymbol{e}_{c_i(s')} \big| s, a \right]. \tag{7}$$

Appendix A outlines the details of this calculation. Consequently, if a function $\boldsymbol{f}_i$ predicts the expected next latent state $\mathbb{E}_p \left[ \boldsymbol{e}_{c_i(s')} \big| s, a \right]$, then Equation (7) can be used to predict the SF vector $\boldsymbol{\psi}_i^\pi(s, a)$. Similar to the

reward-refinement step, a vector-valued function $\boldsymbol{f}_i$ is obtained by solving[1]

$$\boldsymbol{f}_i = \arg\min_{\boldsymbol{f}} \sum_{(s,a,r,s') \in \mathcal{D}} ||\boldsymbol{f}(s,a) - \boldsymbol{e}_{c_i(s')}||. \tag{8}$$

Similar to learning the approximate reward function, we found that it is more practical to train a classifier and to replace the mean squared error loss objective stated in line (8) with a cross entropy loss objective and train the network $\boldsymbol{f}_i$ to predict a probability vector over next latent states. This change is outlined in Algorithm 1. The next clustering $c_{i+1}$ is then constructed such that for two arbitrary states $s$ and $\tilde{s}$,

$$c_{i+1}(s) = c_{i+1}(\tilde{s}) \implies \sum_{a \in \mathcal{A}} ||\hat{\boldsymbol{\psi}}_i^{\pi}(s,a) - \hat{\boldsymbol{\psi}}_i^{\pi}(\tilde{s},a)|| \leq \varepsilon_{\psi}. \tag{9}$$

This SF refinement procedure is repeated until two consecutive clusterings $c_i$ and $c_{i+1}$ are identical.

Algorithm 1 summarizes the outlined method. In the remainder of this section, we will discuss under which assumptions this method computes a reward-predictive representation with as few latent states as possible.

---

**Algorithm 1** Iterative reward-predictive representation learning

1: **Input:** A trajectory data set $\mathcal{D}$, $\varepsilon_r, \varepsilon_{\psi} > 0$.
2: Bin reward values and construct a reward vector $\boldsymbol{w}_r(i) = r_i$.
3: Construct the function $i(r)$ that indexes distinct reward values and $\boldsymbol{w}_r(i(r)) = r$.
4: Solve $\boldsymbol{f}_r = \arg\min_{\boldsymbol{f}} \sum_{(s,a,r,s') \in \mathcal{D}} H(\boldsymbol{f}(s,a), \boldsymbol{e}_{i(r)})$ via gradient descent
5: Compute reward predictions $f_r(s,a) = \boldsymbol{w}_r^{\top} \boldsymbol{f}_r(s,a)$
6: Construct $c_1$ such that $c_1(s) = c_1(\tilde{s}) \implies \sum_{a \in \mathcal{A}} |f_r(s,a) - f_r(\tilde{s},a)| \leq \varepsilon_r$
7: **for** $i = 2, 3, ..., N$ until $c_{i+1} = c_i$ **do**
8:     Compute $\boldsymbol{F}_a$ for every action.
9:     Construct $\boldsymbol{\phi}_i : s \mapsto \boldsymbol{e}_{c_i(s)}$
10:     Solve $\boldsymbol{f}_i = \arg\min_{\boldsymbol{f}} \sum_{(s,a,r,s') \in \mathcal{D}} H(\boldsymbol{f}(s,a), \boldsymbol{e}_{c_i(s')})$ via gradient descent
11:     Compute $\hat{\boldsymbol{\psi}}_i^{\pi}(s,a) = \boldsymbol{e}_{c_i(s)} + \gamma \boldsymbol{F} \boldsymbol{f}_i(s,a)$
12:     Construct $c_{i+1}$ such that $c_{i+1}(s) = \hat{c}_{i+1}(\tilde{s}) \implies \sum_{a \in \mathcal{A}} ||\hat{\boldsymbol{\psi}}_i^{\pi}(s,a) - \hat{\boldsymbol{\psi}}_i^{\pi}(\tilde{s},a)|| \leq \varepsilon_{\psi}$
13: **end for**
14: **return** $\boldsymbol{\phi}_N$

---

### 3.3 Convergence to maximally compressed reward-predictive representations

The idea behind Algorithm 1 is similar to the block-splitting method introduced by Givan et al. (2003). While Givan et al. focus on the tabular setting and refine partitions using transition and reward tables, our clustering algorithm implements a similar refinement method but for data sets sampled from MDPs with perhaps (uncountably) infinite state spaces. Instead of assuming access to the complete transition function, Algorithm 1 learns SFs and uses them to iteratively refine state partitions. For this refinement operation to converge to a correct and maximally-compressed-reward-predictive representation, the algorithm needs to consider all possible transitions between individual state partitions. This operation is implicitly implemented by clustering SFs, which predict the frequency of future state partitions and therefore implicitly encode the partition-to-partition transition table.[2]

Convergence to a correct maximally-compressed-reward-predictive representation relies on two properties that hold at every iteration (please refer to Appendix B for a formal statement of these properties):

1. State partitions are refined and states of different partitions are never merged into the same partition.

2. Two states that lead to equal expected reward sequences are never split into separate partitions.

---

[1] Here, the L2 norm of a vector $\boldsymbol{v}$ is denoted with $||\boldsymbol{v}||$.
[2] The state-to-state transition table is never computed by our algorithm.

The first property ensures that Algorithm 1 is a partition refinement algorithm, as illustrated by the tree schematic in Figure 1(d) (and does not merge state partitions). If such an algorithm is run on a finite trajectory data set with a finite number of state observations, the algorithm is guaranteed to terminate and converge to some state representation because one can always assign every observation into a singleton cluster. However, the second property ensures that the resulting representation is reward-predictive while using as few state partitions as possible: If two states $s$ and $\tilde{s}$ lead to equal expected reward sequences and $\mathbb{E}_p[(r_1, ..., r_n)|s, a_1, ..., a_n] = \mathbb{E}_p[(r_1, ..., r_n)|\tilde{s}, a_1, ..., a_n]$ (for any arbitrary action sequence $a_1, ..., a_n$), then they will not be split into separate partitions. If Algorithm 1 does not terminate early (which we prove in Appendix B), the resulting representation is reward-predictive and uses as few state partitions as possible.

The reward-refinement step satisfies both properties: The first property holds trivially, because $c_1$ is the first partition assignment. The second property holds because two states with different one-step rewards cannot be merged into the same partition for any reward-predictive representation.

To see that both properties are preserved in every subsequent iteration, we consider the partition function $c^*$ of a correct maximally compressed reward-predictive representation. Suppose $c_i$ is a sub-partitioning of $c^*$ and states that are assigned different partitions by $c_i$ are also assigned different partitions in $c^*$. (For example, in Figure 1(d) $c_0$, $c_1$, and $c_3$ are all valid sup-partitions of $c_4$.) Because of this sub-partition property, we can define a projection matrix $\boldsymbol{\Phi}_i$ that associates partitions defined by $c^*$ with partitions defined by $c_i$. Specifically, the entry $\boldsymbol{\Phi}_i(k, j)$ is set to one if for the same state $s$, $c^*(s) = j$ and $c_i(s) = k$. In Appendix B we show that this projection matrix can be used to relate latent states induced by $c^*$ to latent states induced by $c_i$ and

$$\boldsymbol{\Phi}_i \boldsymbol{e}_{c^*(s)} = \boldsymbol{e}_{c_i(s)}. \tag{10}$$

Using the identity in line (10), the SFs at an intermediate refinement iteration can be expressed in terms of the SFs of the optimal reward-predictive representation and

$$\boldsymbol{\psi}_i^\pi(s, a) = \mathbb{E}_{a, \pi}\left[\sum_{t=1}^{\infty} \gamma^{t-1} \boldsymbol{e}_{c_i(s_t)} \middle| s_1 = s, a_1 = a\right] \tag{11}$$

$$= \mathbb{E}_{a, \pi}\left[\sum_{t=1}^{\infty} \gamma^{t-1} \boldsymbol{\Phi}_i \boldsymbol{e}_{c^*(s)} \middle| s_1 = s, a_1 = a\right] \qquad \text{(by substitution with (10))} \tag{12}$$

$$= \boldsymbol{\Phi}_i \mathbb{E}_{a, \pi}\left[\sum_{t=1}^{\infty} \gamma^{t-1} \boldsymbol{e}_{c^*(s)} \middle| s_1 = s, a_1 = a\right] \qquad \text{(by linearity of expectation)} \tag{13}$$

$$= \boldsymbol{\Phi}_i \boldsymbol{\psi}_*^\pi(s, a). \tag{14}$$

As illustrated in Figure 1(c), Lehnert & Littman (2020) showed that two states $s$ and $\tilde{s}$ that are assigned the same partition by a maximally compressed reward-predictive clustering $c^*$ also have equal SF vectors and therefore

$$\boldsymbol{\psi}_i^\pi(s, a) - \boldsymbol{\psi}_i^\pi(\tilde{s}, a) = \boldsymbol{\Phi}_i \boldsymbol{\psi}_*^\pi(s, a) - \boldsymbol{\Phi}_i \boldsymbol{\psi}_*^\pi(\tilde{s}, a) = \boldsymbol{\Phi}_i \underbrace{(\boldsymbol{\psi}_*^\pi(s, a) - \boldsymbol{\psi}_*^\pi(\tilde{s}, a))}_{=\boldsymbol{0}} = \boldsymbol{0}. \tag{15}$$

By line (15), these two states $s$ and $\tilde{s}$ also have equal SFs at any of the refinement iterations in Algorithm 1. Consequently, these two states will not be split into two different partitions (up to some approximation error) and the second property holds.

Similarly, if two states are assigned different partitions, then the first term in the discounted summation in line (11) contains two different one-hot bit vectors leading to different SFs for small enough discount factor and $\varepsilon_\psi$ settings. In fact, in Appendix B we prove that this is the case for all possible transition functions if

$$\gamma < \frac{1}{2} \text{ and } \frac{2}{3}\left(1 - \frac{\gamma}{1 - \gamma}\right) > \varepsilon_\psi > 0. \tag{16}$$

While this property of SFs ensures that Algorithm 1 always refines a given partitioning for any arbitrary transition function, we found that significantly higher discount factor settings can be used in our simulations.

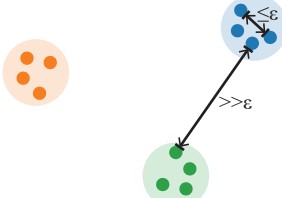

Figure 3: The cluster thresholds $\varepsilon_\psi$ and $\varepsilon_r$ must be picked to account for prediction errors while ensuring that states are not merged into incorrect clusters. For example, suppose the clustered SF vectors are the three black dots in $\mathbb{R}^2$ and the function $\boldsymbol{f}_i$ predicts values close to these dots, as indicated by the colored dots. For the clustering to be correct (and computable in polynomial time), the prediction errors—the distance between the predictions and the correct value—has to be $\varepsilon_\psi/2$. At the same time, $\varepsilon_\psi$ has to be small enough to avoid overlaps between the different coloured clusters.

Because function approximation is used to predict the quantities used for clustering, prediction errors can corrupt this refinement process. If prediction errors are too high, the clustering steps in Algorithm 1 may make incorrect assignments between state observations and partitions. To prevent this, the prediction errors of the learned function $f_r$ and $\boldsymbol{\psi}_i^\pi$ must be bounded by the thresholds used for clustering, leading to the following assumption.

**Assumption 1** ($\varepsilon$-perfect). For $\varepsilon_\psi, \varepsilon_r > 0$, the ERM steps in Algorithm 1 lead to function approximators that are near optimal such that for every observed state-action pair $(s, a)$,

$$\left| f_r(s, a) - \mathbb{E}[r(s, a, s')|s, a] \right| \le \frac{\varepsilon_r}{2} \text{ and } \left|\left| \widehat{\boldsymbol{\psi}}_i^\pi(s, a) - \boldsymbol{\psi}_i^\pi(s, a) \right|\right| \le \frac{\varepsilon_\psi}{2}. \tag{17}$$

Figure 3 illustrates why this assumption is necessary and why predictions have to fall to the correct value in relation to $\varepsilon_\psi$ and $\varepsilon_r$. In Section 4 we will discuss that this assumption is not particularly restrictive in practice and when not adhering to this assumption can still lead to a maximally-compressed-reward-predictive representation. Under Assumption 1, Algorithm 1 converges to a maximally compressed reward-predictive representation.

**Theorem 1** (Convergence). If Assumption 1 and the matching condition in line (16) hold, then Algorithm 1 returns an approximate maximally-compressed-reward-predictive representation for a trajectory data set sampled from any MDP.

A formal proof of Theorem 1 is presented in Appendix B.

In practice, one cannot know if prediction errors are small enough, a principle that is described by Vapnik (1992). However, recent advances in deep learning (Belkin et al., 2019) have found that increasing the capacity of neural networks often makes it possible to interpolate the training data and still perform almost perfectly on independently sampled test data. In the following section we present experiments that illustrate how this algorithm can be used to find a maximally compressed reward-predictive representation.

## 4 Learning reward-predictive representation networks

In this section, we first illustrate how the clustering algorithm computes a reward-predictive representation on the didactic Column World example. Then, we focus on a more complex visual control task—the Combination Lock task, where inputs are a set of MNIST images from pixels—and discuss how function approximation errors lead to spurious latent states and how they can be filtered out. Lastly, we present a set of experiments highlighting how initializing a DQN agent with a reward-predictive representation network

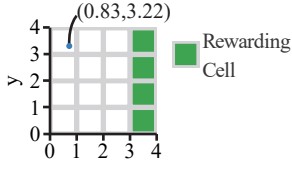
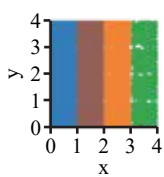
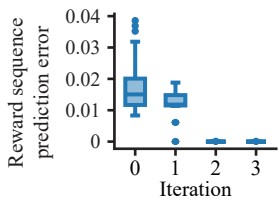

(a) Point Observation Column World task  (b) Reward-predictive clustering  (c) Reward sequence prediction errors

Figure 4: Reward-predictive clustering of the Point Observation Column World task. (a): The Point Observation Column World task is a variant of the Column World task where instead of providing the agent with a grid cell index it only observes a real valued point $(x, y) \in (0, 4)^2$. When the agent is in a grid cell, for example cell the top left cell, a point is sampled uniformly at random from the corresponding cell, for example the point $(0.83, 3.22)$. (b): The computed cluster function $c_3$ assigns each state observation (a point in the shown scatter plot) with a different latent state index (a different color). (c): The box plot shows the reward sequence prediction error for each trajectory at each iteration (iteration 0 shows the initial cluster function). At each iteration a different representation network was trained and then evaluated on a separately sampled 100-trajectory test data set. The full details of this experiment are listed in Appendix C.

improves learning efficiency, demonstrating in which cases reward-predictive representations are suitable for out-of-distribution generalization.

Figure 4 illustrates a reward-predictive clustering for a variant of the Column World task where state observations are real-valued points. This variant is a block MDP (Du et al., 2019): Instead of observing a grid cell index, the agent observes a real-valued point $(x, y)$ (Figure 4(a)) but still transitions through a $4 \times 4$ grid. This point is sampled uniformly at random from a square that corresponds to the grid cell the agent is in, as illustrated in Figure 4(a). Therefore, the agent does not (theoretically) observe the same $(x, y)$ point twice and transitions between different states become probabilistic. For this task, a two-layer perceptron was used to train a reward and next latent state classifier (Algorithm 1, lines 4 and 10). Figure 4(b) illustrates the resulting clustering as colouring of a scatter plot. Each dot in the scatter plot corresponds to a state observation point $(x, y)$ in the training data set and the colouring denotes the final latent state assignment $c_3$. Figure 4(c) presents a box-plot of the reward-sequence prediction errors as a function of each refinement iteration. One can observe that after performing the second refinement step and computing the cluster function $c_2$, all reward-sequence prediction errors drop to zero. This is because the clustering algorithm initializes the cluster function $c_0$ by first merging all terminal states into a separate partition (and our implementation of the clustering algorithm is initialized at the second step in Figure 1(d)). Because cluster functions $c_2$ and $c_3$ are identical in this example, the algorithm is terminated after the third iteration.

## 4.1 Clustering with function approximation errors

As illustrated in Figure 3, for the cluster algorithm to converge to a maximally compressed representation, the predictions made by the neural networks must be within some $\varepsilon$ of the true prediction target. Depending on the task and training data set, this objective may be difficult to satisfy. Belkin et al. (2019) presented the double-descent curve, which suggests that it is possible to accurately approximate any function with large enough neural network architectures. In this section we test the assumption that all predictions must be $\varepsilon$ accurate by running the clustering algorithm on a data set sampled from the Combination Lock task (Figure 5). In this task, the agent decides which dial to rotate on each step to unlock a numbered combination lock (schematic in Figure 5(a)). Here, state observations are assembled using training images from the MNIST data set (Lecun et al., 1998) and display three digits visualizing the current number combination of the lock. To compute a reward-predictive representation for this task, we adapt our clustering algorithm to process images using the ResNet18 architecture (Paszke et al., 2019; He et al., 2016) for approximating one-step rewards and next latent states. For all experiments we initialize all network weights randomly and do not provide any pre-trained weights. The full details of this experiment are documented in Appendix C.

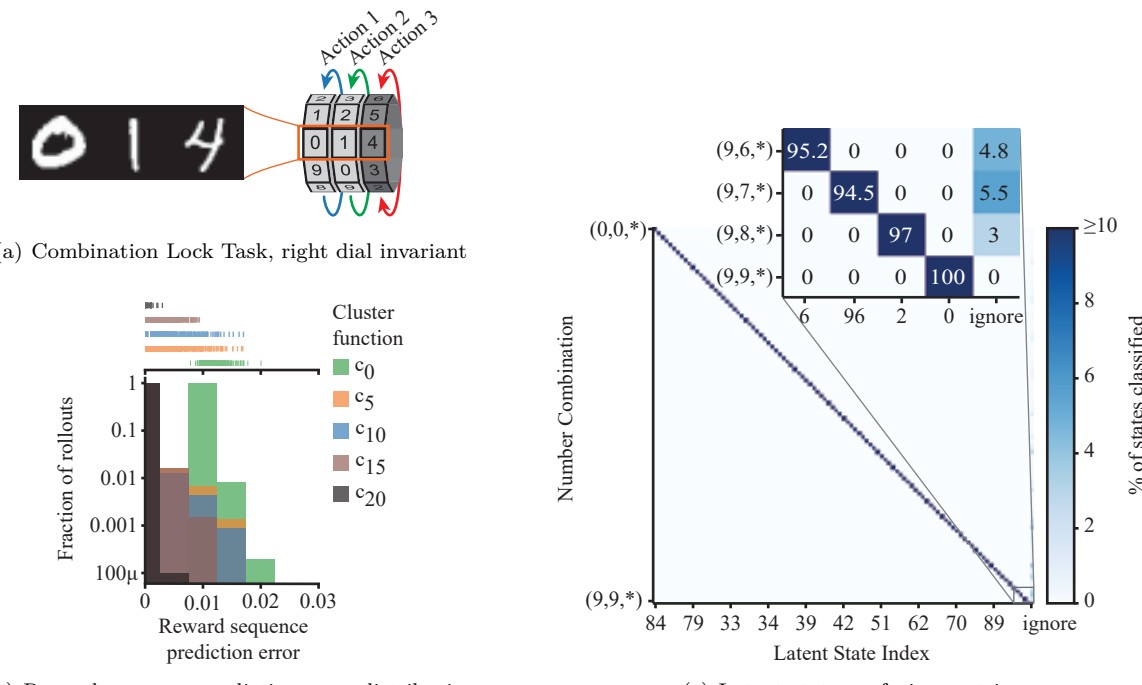

(a) Combination Lock Task, right dial invariant

(b) Reward sequence prediction error distribution

(c) Latent state confusion matrix

Figure 5: Reward-predictive clustering of the Combination Lock task. (a): In the Combination Lock task, the agent decides which dial(s) to rotate to move toward a rewarding combination. The agent has to learn that only the first two dials are relevant for unlocking the combination: a reward is given once the left and center dials both arrive at the digit nine and the lock matches the pattern $(9, 9, *)$. The right (shaded) dial is "broken" and spins at random when the third action is selected, and thus all digits on it should be equally reward-predictive. Each state consists of an image that is assembled using the MNIST data set. The fixed trajectory data set provided to the clustering algorithm uses images from the MNIST training dataset. The resulting model was evaluated using an independently sampled test trajectory data set using images from the MNIST test data set. (b): The histogram plots the distribution reward sequence errors for 1000 test trajectories for five different refinement stages of the clustering algorithm on a log-scale. The distribution of the 1000 samples is plotted as a rug plot above the histogram. For each trajectory the absolute difference between predicted and true reward value was computed and averaged along the trajectory. The predictions where made by training a separate representation network for each cluster function. (c): Matrix plot illustrating how different number combinations are associated with different latent states. Each row plots the distribution across latent states of images matching a specific number pattern. Each column of the matrix plot corresponds to a specific latent state index and which combination is associated with which index is determined arbitrarily by the clustering algorithm. Terminal states that are observed at the end of each trajectory are merged into latent state zero by default. The ignore column indicates the fraction of state images that were identified as belonging to a spurious latent state and are excluded from the final clustering.

In this task, a reward-predictive representation network has to not only generalize across variations in individual digits, but also learn to ignore the rightmost digit. The matrix plot in Figure 5(c) illustrates how the reward-predictive representation network learned by the clustering algorithm generalizes across the different state observations. Intuitively, this plot is similar to a confusion matrix: Each row plots the distribution over latent states for all images that match a specific combination pattern. For example, the first row plots the latent state distribution for all images that match the pattern $(0, 0, *)$ (left and middle dial are set to zero, the right dial can be any digit), the second row plots the distribution for the pattern $(0, 1, *)$, and so on. In total the clustering algorithm correctly inferred 100 reward-predictive latent states and correctly ignores the rightmost digit, abstracting it away from the state input. Prediction errors can contort the clustering in two ways:

1. If prediction errors are high, then a state observation can be associated with the wrong latent state. For example, an image with combination $(0, 1, 4)$ could be associated with the latent state corresponding to the pattern $(0, 7, *)$.

2. If prediction errors are low but still larger than the threshold $\varepsilon_\psi$ or $\varepsilon_r$, then some predictions can be assigned into their own cluster and a spurious latent state is created. These spurious states appear as latent states that are associated with a small number of state observations.

Figure 5(c) indicates that the first prediction error type does not occur because all off-diagonal elements are exactly zero. This is because a large enough network architecture is trained to a high enough accuracy. However, the second prediction error type does occur. In this case, latent states that are associated with very few state observations are masked out of the data set used for training the neural network (line 10 in Algorithm 1). These states are plotted in the ignore column (right-most column) in Figure 5(c). In total, less than 0.5% of the data set are withheld and the clustering algorithm has inferred 100 latent states. Consequently, the learned reward-predictive representation uses as few latent states as possible and is maximally compressed.

Figure 5(b) plots the reward-sequence error distribution for a representation network at different refinement stages. Here, 1000 independently sampled test trajectories were generated using images from the MNIST test set. One can see that initially reward sequence prediction errors are high and then converge towards zero as the refinement algorithm progresses. Finally, almost all reward sequences are predicted accurately but not perfectly, because a distinct test image set is used and the representation network occasionally predicts an incorrect latent state. This is a failure in the vision model—if the convolutional neural network would perfectly classify images into the latent states extracted by the clustering algorithm, then the reward sequence prediction errors would be exactly zero (similar to the Column World example in Figure 4(c)). Furthermore, if the first transition of a 1000-step roll-out is incorrectly predicted, then all subsequent predictions are incorrect as well. Consequently, the reward sequence prediction error measure is sensitive to any prediction errors that may happen when predicting rewards for a long action sequence. However, the trend of minimizing reward sequence prediction errors with every refinement iteration is still plainly visible in Figure 5(b).

## 4.2 Improving learning efficiency

Ultimately, the goal of using reward-predictive representations is to speed up learning by re-using abstract task knowledge encoded by a pre-trained representation network. In contrast, established meta-learning algorithms such as MAML (Finn et al., 2017) or the SF-based Generalized Policy Improvement (GPI) algorithm (Barreto et al., 2018; 2020) rely on extracting either one or multiple network initializations to accelerate learning in a test task. To empirically test the differences between re-using a pre-trained reward-predictive representation network and using a previously learned network initialization, we now consider three variants of the Combination Lock task (Figure 6(a)). All variants vary from the training task in their specific transitions, rewards, and optimal policy. Furthermore, the state images are generated using MNIST test images to test if a pre-trained agent can generalize what it has seen during pre-training to previously unseen variations of digits.[3] The three task variants require an agent to process the state images differently

---

[3]This experiment design is similar to using separately sampled training and test data in supervised machine learning.

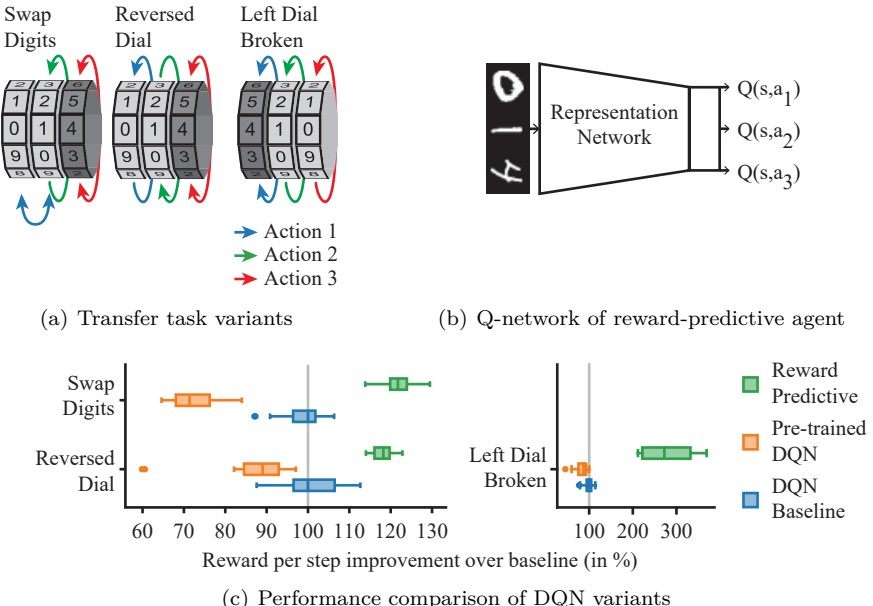

(a) Transfer task variants

(b) Q-network of reward-predictive agent

(c) Performance comparison of DQN variants

Figure 6: Representation transfer in the Combination Lock task. (a): In the swap digits variant, the transition function is changed such that the first action only swaps the digit between the left and middle dial. Only the middle dial rotates as before and the right dial also does not have any effect on the obtained rewards. Furthermore, the rewarding combination is changed to $(5, 6, *)$. The reversed dial variant differs from the training task in that the rotation direction of the middle dial is reversed and the rewarding combination is changed to $(7, 4, *)$. The left dial broken variant is similar to the training task but the left dial is broken and spins at random instead of the right dial. Here, the transitions and reward association between different latent states are the same as in the training task with the difference being how different images are associated with different latent states and different action labels having different effects. The rewarding combination is $(*, 9, 9)$. To ensure that the state images of the test tasks are distinct from the training task, all test tasks construct the state images using the MNIST test image set. (b): The reward-predictive agent replaces all except the top-most layer with the reward-predictive representation network computed by the clustering algorithm for the training task. During training in the test task only the top-most layer receives gradient updates and the representation network's weights are not changed. (c): Each agent was trained for 20 different seeds in each task. For each repeat, the pre-trained DQN agent was first trained on the training task and then on the test task. Appendix C lists all details and additional plots of the experiment.

in order to maximize rewards: In the swap digits and reversed dial variants (center and left schematic in Figure 6(a)), an agent has to correctly recognize the left and center digit in order to select actions optimally. While the effect of different actions and the rewarding combinations differ from the training task, an agent initially processes state images in the same way as in the training task. Specifically, because the right dial is still broken and rotates at random, an agent needs to correctly identify the left and center digits and then use that information to make a decision. These two transfer tasks test an agent's ability to adapt to different transitions and rewards while preserving which aspects of the state image—namely the left and center digits—are relevant for decision-making. The left dial broken variant (right schematic in Figure 6) differs in this particular aspect. Here, the center and right digits are relevant for reward-sequence prediction and decision-making because the left dial is broken and rotates at random. With this task, we test to what extent a pre-trained reward-predictive representation network can be used when state equivalences modelled by the representation network differ between training and test tasks.

To test for positive transfer in a controlled experiment, we train three variants of the DQN algorithm (Mnih et al., 2015) and record the average reward per time step spent in each task. Each DQN variant uses a different Q-network initialisation but all agents use the same network architecture, number of network weights, and hyper-parameters. Hyper-parameters were independently fine tuned on the training task in Figure 5(a) so as to not bias the hyper-parameter selection towards the used test tasks (and implicitly using information about the test tasks during training). In Figure 6(c), the DQN baseline (shown in blue) initializes networks at random (using Glorot initialization (Glorot & Bengio, 2010)) similar to the original DQN agent. This agent's performance is used as a reference value in each task. The pre-trained DQN agent (shown in orange) first learns to solve the training task, and the learned Q-network weights are then used to initialize the network weights in each test task. By pre-training the Q-network in this way, the DQN agent has to adapt the previously learned solution to the test task. Here, the pre-trained DQN agent initially repeats the previously learned behaviour—which is not optimal in any of the test tasks—and then has to re-learn the optimal policy for each test task. This re-learning seems to negatively impact the overall performance of the agent and it would be more efficient to randomly initialize the network weights (Figure 6(c)).

This approach of adapting a pre-trained Q-network to a test task is used by both MAML and SF-based GPI. While these methods rely on extracting information from multiple training tasks, the results in Figure 6(c) demonstrate that if training and test tasks differ sufficiently, then re-using a pre-trained Q-network to initialize learning may negatively impact performance and a new Q-network or policy may have to be learned from scratch (Nemecek & Parr, 2021). Reward-predictive representations enable a more abstract form of task knowledge re-use that is more robust in this case. This is illustrated by the reward-predictive agent in Figure 6(c) that outperforms the other two agents. The reward-predictive agent (shown in green in Figure 6(c)) sets all weights except for the top-most linear layer to the weights of the reward-predictive representation network learned by the clustering algorithm for the training task (Figure 6(b)). Furthermore, no weight updates are performed on the representation network itself—only the weights of the top-most linear layer are updated during learning in the test task. By re-using the pre-trained representation network, the reward-predictive agent maps all state images into one of the 100 pre-trained latent states resulting in a significant performance improvement. This performance improvement constitutes a form of systematic out-of-distribution generalization, because the reward-predictive representation network is not adjusted during training and because trajectories observed when interacting with the test task are out-of-distribution of the trajectories observed during pre-training.

Interestingly, in the left dial broken variant the performance improvement of the reward-predictive agent is even more significant. This result is unexpected, because in this case the state equivalences modelled by the transferred representation function differ between the training and the test tasks: In the training task, the right dial is irrelevant for decision-making and can be abstracted away whereas in the test task the left dial is irrelevant for decision-making and can be abstracted away instead. Consequently, a representation that is reward-predictive in the training task is not reward-predictive in the left dial broken test task and an RL agent would have to re-train a previously learned representation for it be reward predictive in the test task. Nevertheless, the reward-predictive representation network can still be used to maximize rewards in this task variant: The agent first learns to rotate the center dial to the rewarding digit "9". This is possible because the network can still leverage parts of the reward-predictive abstraction that remain useful

for the new task. In this case, the center digits are still important as they were in the original task and the reward-predictive representation network maps distinct center digits to distinct latent states, although the combination $(1, 9, *)$ and $(2, 9, *)$ are mapped to different latent states given the representation learned in the training task. Once the center dial is set to the digit "9", the agent can simply learn a high Q-value for the action associated with rotating the third dial, and it does so until the rewarding combination is received. Because the reward predictive agent is a variant of DQN and initializes Q-values to be close to zero, the moment the algorithm increases a Q-value through a temporal-difference update, the agent keeps repeating this action with every greedy action selection step and does not explore all possible states, resulting in a significant performance improvement.[4] While the reward-predictive representation network cannot be used to predict reward-sequences or event Q-values accurately, the Q-value predictions learned by the agent are sufficient to still find an optimal policy quickly in this test task. Of course, one could imagine test tasks where this is not the case and the agent would have to learn a new policy from scratch.

This experiment highlights how reward-predictive representation networks can be used for systematic out-of-distribution generalization. Because the representation network only encodes state equivalences, the network can be used across tasks with different transitions and rewards. However, if different state equivalences are necessary for reward prediction in a test task, then it may or may not be possible to learn an optimal policy without modifying the representation network. The left dial broken test task in Figure 5 presents a case where state equivalences differ from the training task but it is still possible to accelerate learning of an optimal policy significantly.

## 5 Discussion

In this article, we present a clustering algorithm to compute reward-predictive representations that use as few latent states as possible. Unlike prior work (Lehnert & Littman, 2020; 2018), which learns reward-predictive representations through end-to-end gradient descent, our approach is similar to the block splitting method presented by Givan et al. (2003) for learning which two states are bisimilar in an MDP. By starting with a single latent state and then iteratively introducing additional latent states to minimize SF prediction errors where necessary, the final number of latent states is minimized. Intuitively, this refinement is similar to temporal-difference learning, where values are first updated where rewards occur and subsequently value updates are bootstrapped at other states. The clustering algorithm computes a reward-predictive representation in a similar way, by first refining a state representation around changes in one-step rewards and subsequently bootstrapping from this representation to further refine the state clustering. This leads to a maximally compressed latent state space, which is important for abstracting away information from the state input and enabling an agent to efficiently generalize across states (as demonstrated by the generalization experiments in Section 4.2). Such latent state space compression cannot be accomplished by auto-encoder based architectures (Ha & Schmidhuber, 2018) or frame prediction architectures (Oh et al., 2015; Leibfried et al., 2016; Weber et al., 2017) because a decoder network requires the latent state to be predictive of the entire task state. Therefore, these methods encode the entire task state in a latent state without abstracting any part of the task state information away.

Prior work (Ferns et al., 2004; Comanici et al., 2015; Gelada et al., 2019; Zhang et al., 2021b;a) has focused on using the Wasserstein metric to measure how bisimilar two states are. Computing the Wasserstein metric between two states is often difficult in practice, because it requires solving an optimization problem for every distance calculation and it assumes a measurable state space—an assumption that is difficult to satisfy when working with visual control tasks for example. Here, approximations of the Wasserstein metric are often used but these methods introduce other assumptions instead, such as a normally distributed next latent states (Zhang et al., 2021a) or a Lipschitz continuous transition function where the Lipschitz factor is $1/\gamma$ (Gelada et al., 2019)[5]. The presented refinement method does not require such assumptions, because the presented algorithm directly clusters one-step rewards and SFs for arbitrary transition and reward functions. SFs, which encode the frequencies of future states, provide a different avenue to computing which two states are bisimilar without requiring a distance function on probability distributions such as the Wasserstein

---

[4]For all experiments we use a $\varepsilon$-greedy action selection strategy that initially selects actions uniformly at random but becomes greedy with respect to the predicted Q-values within the first 10 episodes.

[5]Here, $\gamma \in (0, 1)$ is the discount factor.

metric. Nonetheless, using the Wasserstein metric to determine state bisimilarity may provide an avenue for over-compressing the latent state space at the expense of increasing prediction errors (Ferns et al., 2004; Comanici et al., 2015) (for example, compressing the Combination Lock task into 90 latent states instead of 100).

A key challenge in scaling model-based RL algorithms is the fact that these agents are evaluated on their predictive performance. Consequently, any approximation errors (caused by not adhering to the $\varepsilon$-perfection assumption illustrated in Figure 3) impact the resulting model's predictive performance—a property common to model-based RL algorithms (Talvitie, 2017; 2018; Asadi et al., 2018). Evaluating a model's predictive performance is more stringent than what is typically used for model-free RL algorithms such as DQN. Typically, model-free RL algorithms are evaluated on the learned optimal policy's performance and are not evaluated on their predictive performance. For example, while DQN can learn an optimal policy for a task, the learned Q-network's prediction errors may still be high for some inputs (Witty et al., 2018). Prediction errors of this type are often tolerated, because model-free RL algorithms are benchmarked based on the learned policy's ability to maximize rewards and not their accuracy of predicting quantities such as Q-values or rewards. This is the case for most existing deep RL algorithms that are effectively model-based and model-free hybrid architectures (Oh et al., 2017; Silver et al., 2017a; Gelada et al., 2019; Schrittwieser et al., 2019; Zhang et al., 2021a)—these models predict reward-sequences only over very short horizons (for example, Oh et al. (2017) use 10 time steps). In contrast, reward-predictive representations are evaluated for their prediction accuracy. To achieve low prediction errors, the presented results suggest that finding $\varepsilon$-perfect approximations becomes important. Furthermore, the simulations on the MNIST combination-lock task demonstrate that this goal can be accomplished by using a larger neural network architecture.

To compute a maximally compressed representation, the presented clustering algorithm needs to have access to the entire trajectory training data set at once. How to implement this algorithm in an online learning setting—a setting where the agent observes the different transitions and rewards of a task as a data stream— is not clear at this point. To implement an online learning algorithm, an agent would need to assign incoming state observations to already existing state partitions. Without such an operation it would not be possible to compute a reward-predictive representation that still abstracts away certain aspects from the state itself. Because the presented clustering method is based on the idea of refining state partitions, it is currently difficult to design an online learning agent that does not always re-run the full clustering algorithm on the history of all transitions the agent observed.

One assumption made in the presented experiments is that a task's state space can always be compressed into a small enough finite latent space. This assumption is not restrictive, because any (discrete time) RL agent only observes a finite number of transitions and states at any given time point. Consequently, all state observations can always be compressed into a finite number of latent states, similar to block MDPs (Du et al., 2019). Furthermore, the presented method always learns a fully conjunctive representation. In the combination-lock examples, the reward-predictive representation associates a different latent state (one-hot vector) with each relevant combination pattern. This representation is conjunctive because it does not model the fact that the dials rotate independently. A disjunctive or factored representation could map each of the three dials independently into three separate latent state vectors and a concatenation of these vectors could be used to describe the task's latent state. Such a latent representation is similar to factored representations used in prior work (Guestrin et al., 2003; Diuk et al., 2008) and these factored representations permit a more compositional form of generalization across different tasks (Kansky et al., 2017; Battaglia et al., 2016; Chang et al., 2016). How to extract such factored representations from unstructured state spaces such as images still remains a challenging problem. We leave such an extension to future work.

Prior work on (Deep) SF transfer (Barreto et al., 2018; 2020; Kulkarni et al., 2016; Zhang et al., 2017), meta-learning (Finn et al., 2017), or multi-task learning (Rusu et al., 2015; D'Eramo et al., 2020) has focused on extracting an inductive bias from a set of tasks to accelerate learning in subsequent tasks. These methods transfer a value function or policy model to initialize and accelerate learning. Because these methods transfer a model of a task's policy, these models have to be adapted to each transfer task, if the transfer task's optimal policy differs from the previously learned policies. Reward-predictive representations overcome this limitation by only modelling how to generalize across different states. Because reward-predictive representations do not encode the specifics of how to transition between different latent states or how latent states are tied to

rewards, these representations are robust to changes in transitions and rewards. Furthermore, the reward-predictive representation network is learned using a single task and the resulting network is sufficient to demonstrate positive transfer across different transitions and rewards. This form of transfer is also different from the method presented by Zhang et al. (2021b), where the focus is on extracting a common task structure from a set of tasks instead of learning a representation from a single task and transferring it to different test tasks. Still, in a lifelong learning scenario, re-using the same reward-predictive representation network to solve every task may not be possible because an agent may have to generalize across different states (as demonstrated by the left dial broken combination lock variant in Section 4.2). In this article, we analyze the generalization properties of reward-predictive representations through A-B transfer experiments. While Lehnert et al. (2020) already present a (non-parametric) meta-learning model that uses reward-predictive representations to accelerate learning in finite MDPs, we leave how to integrate the presented clustering algorithm into existing meta-learning frameworks commonly used in deep RL—such as Barreto et al. (2018) or Finn et al. (2017)—for future work.

## 6 Conclusion

We presented a clustering algorithm to compute reward-predictive representations that introduces as few latent states as possible. The algorithm works by iteratively refining a state representation using a temporal difference error that is defined on state features. Furthermore, we analyze under which assumptions the resulting representation networks are suitable for systematic out-of-distribution generalization and demonstrate that reward-predictive representation networks enable RL agents to re-use abstract task knowledge to improve their learning efficiency.

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

## Appendix A    Linear Successor Feature Models

Lehnert & Littman define LSFMs as a set of real-valued vectors $\{\boldsymbol{w}_a\}_{a\in\mathcal{A}}$ and real-valued square matrices $\{\boldsymbol{F}_a\}_{a\in\mathcal{A}}$ that are indexed by the different actions $a \in \mathcal{M}$ of an MDP. Furthermore, LSFMs can be used to identify a reward-predictive representation function $\boldsymbol{\phi} : \mathcal{S} \to \mathbb{R}^n$. Specifically, if a state-representation function $\boldsymbol{\phi}$ satisfies for all state-action pairs $(s, a)$

$$\boldsymbol{w}_a^\top \boldsymbol{\phi}(s) = \mathbb{E}_p[r(s, a, s')|s, a] \tag{18}$$

$$\text{and } \boldsymbol{F}_a^\top \boldsymbol{\phi}(s) = \boldsymbol{\phi}(s) + \gamma \overline{\boldsymbol{F}}^\top \mathbb{E}_p[\boldsymbol{\phi}(s')|s, a] \text{ where } \overline{\boldsymbol{F}} = \frac{1}{|\mathcal{A}|}\sum_{a'\in\mathcal{A}} \boldsymbol{F}_{a'}, \tag{19}$$

then the state-representation function $\boldsymbol{\phi}$ is reward-predictive.

Given a partition function $c$ and the trajectory data set $\mathcal{D}$, a LSFM can be computed. For a partition $i$ the $i$th entry of the weight vector $\boldsymbol{w}_a$ equals the one-step rewards averaged across all state observations and

$$\boldsymbol{w}_a(i) = \frac{1}{|\{(s, a, r, s')|c(s) = i\}|} \sum_{(s,a,r,s')|c(s)=i} r, \tag{20}$$

where the summation Equation 20 ranges over all transitions in $\mathcal{D}$ that start in partition $i$. Similarly, the empirical partition-to-partition transition probabilities can be calculated and stored in a row-stochastic transition matrix $\boldsymbol{M}_a$. Each entry of this matrix is set to the empirical probability of transitioning from a partition $i$ to a partition $j$ and

$$\boldsymbol{M}_a(i, j) = \frac{|\{(s, a, r, s')|c(s) = i, c(s') = j\}|}{|\{(s, a, r, s')|c(s) = i\}|}. \tag{21}$$

Using this partition-to-partition transition matrix, the matrices $\{\boldsymbol{F}_a\}_{a\in\mathcal{A}}$ can be calculated as outlined by Lehnert & Littman and

$$\boldsymbol{F}_a = \boldsymbol{I} + \gamma \boldsymbol{M}_a \boldsymbol{F} \text{ and } \boldsymbol{F} = (\boldsymbol{I} - \gamma \overline{\boldsymbol{M}})^{-1}, \tag{22}$$

where $\boldsymbol{M} = \frac{1}{|\mathcal{A}|}\sum_{a\in\mathcal{A}} \boldsymbol{M}_a$.

This calculation is used to compute the SF targets used for function approximation in Algorithm 1.

## Appendix B    Convergence proof

**Definition 1** (Sub-clustering). A clustering $c$ is a sub-clustering of $c^*$ if the following property holds:

$$\forall s, \tilde{s}, \ c(s) \neq c(\tilde{s}) \implies c^*(s) \neq c^*(\tilde{s}). \tag{23}$$

**Definition 2** (Maximally-Compressed-Reward-Predictive Clustering). A maximally-compressed-reward-predictive representation is a function $c^*$ assigning every state $s \in \mathcal{S}$ to an index such that for all state-action pairs $(s, a)$

$$\left|\boldsymbol{w}_a^\top \boldsymbol{e}_{c*(s)} - \mathbb{E}_p[r(s, a, s')|s, a]\right| \leq \varepsilon_r \tag{24}$$

$$\text{and } \left|\boldsymbol{F}_a^\top \boldsymbol{e}_{c*(s)} - \boldsymbol{\psi}_*^\pi(s, a)\right| \leq \varepsilon_\psi, \tag{25}$$

where $\boldsymbol{\psi}_*^\pi(s, a)$ are the SFs calculated for a state-representation function mapping a state $s$ to a one-hot bit vector $c^*(s)$. Furthermore, this representation uses as few indices as possible.

Definition 2 implicitly makes the assumption that the state space of an arbitrary MDP can be partitioned into finitely many reward-predictive partitions. While this may not be the case for all possible MDPs, this assumption is not restrictive when using the presented clustering algorithm. Because the trajectory data set is finite, any algorithm only processes a finite subset of all possible states (even if state spaces are uncountable infinite) and therefore can always partition these state observations into a finite number of partitions.

**Property 1** (Refinement Property). In Algorithm 1, every iteration refines the existing partitions until the termination condition is reached. Specifically, for every iteration $c_i$ is a sub-clustering of $c_{i+1}$ and for any two distinct states $s$ and $\tilde{s}$,

$$c_i(s) \neq c_i(\tilde{s}) \implies c_{i+1}(s) \neq c_{i+1}(\tilde{s}). \tag{26}$$

**Property 2** (Reward-predictive Splitting Property). Consider a maximally-compressed-reward-predictive representation encoded by the clustering $c^*$ and the cluster sequence $c_1, c_2, \dots$ generated by Algorithm 1. For any two distinct states $s$ and $\tilde{s}$,

$$c_i(s) \neq c_i(\tilde{s}) \implies c^*(s) \neq c^*(\tilde{s}) \tag{27}$$

**Lemma 1** (SF Separation). For a cluster function $c_i$ and any arbitrary MDP, if

$$\gamma < \frac{1}{2} \text{ and } \frac{2}{3}\left(1 - \frac{\gamma}{1-\gamma}\right) > \varepsilon_\psi > 0, \tag{28}$$

then

$$\|\boldsymbol{\psi}_i^\pi(s,a) - \boldsymbol{\psi}_i^\pi(\tilde{s},a)\| \geq 3\varepsilon_\psi \tag{29}$$

for two states $s$ and $\tilde{s}$ that are assigned to two different partitions and $c_i(s) \neq c_i(\tilde{s})$.

*Proof of SF Separation Lemma 1.* First, we observe that the norm of a SF vector can be bounded with

$$\left\|\boldsymbol{\psi}_i^\pi(s,a)\right\| = \left\|\mathbb{E}_\pi\left[\sum_{t=1}^\infty \gamma^{t-1}\boldsymbol{e}_{c_i(s_t)}\Big| s = s_1, a\right]\right\| \tag{30}$$

$$= \left\|\sum_{t=1}^\infty \gamma^{t-1}\mathbb{E}_\pi\left[\boldsymbol{e}_{c_i(s_t)}\Big| s = s_1, a\right]\right\| \qquad \text{(by linearity of expectation)} \tag{31}$$

$$\leq \sum_{t=1}^\infty \gamma^{t-1}\underbrace{\left\|\mathbb{E}_\pi\left[\boldsymbol{e}_{c_i(s_t)}\Big| s = s_1, a\right]\right\|}_{\leq 1} \tag{32}$$

$$= \sum_{t=1}^\infty \gamma^{t-1} \tag{33}$$

$$= \frac{1}{1-\gamma}. \tag{34}$$

The transformation to line (33) uses the fact that expected values of one-hot vectors are always probability vectors.

Furthermore, we note that

$$0 \leq \gamma < \frac{1}{2} \implies \frac{2\gamma}{1-\gamma} < 2. \tag{35}$$

The norm of the difference of SF vectors for two states $s$ and $\tilde{s}$ that start in different partitions can be bounded with

$$\left\|\boldsymbol{\psi}_i^\pi(s,a) - \boldsymbol{\psi}_i^\pi(\tilde{s},a)\right\| = \left\|(\boldsymbol{e}_k + \gamma\mathbb{E}[\boldsymbol{\psi}_i^\pi(s',a')|s,a]) - (\boldsymbol{e}_l + \gamma\mathbb{E}[\boldsymbol{\psi}_i^\pi(s',a')|\tilde{s},a])\right\| \tag{36}$$

$$= \left\|(\boldsymbol{e}_k - \boldsymbol{e}_l) + \gamma(\mathbb{E}[\boldsymbol{\psi}_i^\pi(s',a')|s,a] - \mathbb{E}[\boldsymbol{\psi}_i^\pi(s',a')|\tilde{s},a])\right\| \tag{37}$$

$$= \left\|(\boldsymbol{e}_k - \boldsymbol{e}_l) - \gamma(\mathbb{E}[\boldsymbol{\psi}_i^\pi(s',a')|\tilde{s},a] - \mathbb{E}[\boldsymbol{\psi}_i^\pi(s',a')|s,a])\right\| \tag{38}$$

$$\geq \left|\underbrace{\left\|\boldsymbol{e}_k - \boldsymbol{e}_l\right\|}_{=2} - \gamma\left\|\mathbb{E}[\boldsymbol{\psi}_i^\pi(s',a')|\tilde{s},a] - \mathbb{E}[\boldsymbol{\psi}_i^\pi(s',a')|s,a]\right\|\right| \tag{39}$$

$$= \left|2 - \underbrace{\gamma\left\|\mathbb{E}[\boldsymbol{\psi}_i^\pi(s',a')|\tilde{s},a] - \mathbb{E}[\boldsymbol{\psi}_i^\pi(s',a')|s,a]\right\|}_{\in [0, \frac{2\gamma}{1-\gamma}] \text{ by (34) and} < 2 \text{ by (35)}}\right| \tag{40}$$

$$= 2 - \gamma\left\|\mathbb{E}[\boldsymbol{\psi}_i^\pi(s',a')|\tilde{s},a] - \mathbb{E}[\boldsymbol{\psi}_i^\pi(s',a')|s,a]\right\| \tag{41}$$

$$\geq 2 - \frac{2\gamma}{1-\gamma} \tag{42}$$

The transformation to line (40) holds because $s$ and $\tilde{s}$ start in different partitions and therefore $c_i(s) = k \neq c_i(\tilde{s}) = l$. The transformation to line (41) holds, because the norm of the difference of two SF vectors is bounded by $\frac{2}{1-\gamma}$. The term inside the absolute value calculation cannot possibly become negative because the discount factor $\gamma$ is set to be below $\frac{1}{2}$ and the bound in line (35) holds.

Using the condition on the discount factor in line (28), we have

$$\frac{2}{3}\left(1 - \frac{\gamma}{1-\gamma}\right) \geq \varepsilon_\psi \implies 2 - \frac{2\gamma}{1-\gamma} \geq 3\varepsilon_\psi \qquad \text{(by (28))} \qquad (43)$$

$$\implies ||\boldsymbol{\psi}_i^\pi(s,a) - \boldsymbol{\psi}_i^\pi(\tilde{s},a)|| \geq 3\varepsilon_\psi. \qquad \text{(by (42))} \qquad (44)$$

$\square$

**Definition 3** (Representation Projection Matrix)**.** For a maximally-compressed-reward-predictive clustering $c^*$ and a sub-clustering $c_i$, we define a projection matrix $\boldsymbol{\Phi}_i$ such that every entry

$$\boldsymbol{\Phi}_i(k,l) = \begin{cases} 1 & \exists s \text{ such that } c_i(s) = k \text{ and } c^*(s) = l \\ 0 & \text{otherwise.} \end{cases} \qquad (45)$$

**Lemma 2** (SF Projection)**.** For every state-action pair $(s,a)$, $\boldsymbol{\psi}_i^\pi(s,a) = \boldsymbol{\Phi}_i \boldsymbol{\psi}_*^\pi(s,a)$.

*Proof of SF Projection Lemma 2.* The proof is by the derivation in lines (11) through (14). $\square$

*Proof of Convergence Theorem 1.* The convergence proof argues by induction on the number of refinement iterations and first establishes that the Refinement Property 1 and Reward-predictive Splitting Property 2 hold at every iteration. Then we provide an argument that the returned cluster function is a maximally-compressed-reward-predictive representation.

**Base case:** The first clustering $c_1$ merges two state observations into the same cluster if they lead to equal one-step rewards for every action. The reward-condition in Equation (24) can be satisfied by constructing a vector $\boldsymbol{w}_a$ such that every entry equals the average predicted one-step reward for each partition and

$$\boldsymbol{w}_a(i) = \frac{1}{|\{s : c_1(s) = i\}|} \sum_{s:c_1(s)=i} f_r(s,a) \qquad (46)$$

By Assumption 1, all predictions made by $f_r$ are at most $\frac{\varepsilon_r}{2}$ apart from the correct value and therefore

$$|\boldsymbol{e}_{c_1(s)}^\top \boldsymbol{w}_a - \mathbb{E}_p[r(s,a,s')|s,a]| \leq \varepsilon_r \qquad (47)$$

Consequently, the reward condition in Equation (24) is met and for any two states $s$ and $\tilde{s}$

$$c_1(s) \neq c_1(\tilde{s}) \implies c^*(s) \neq c^*(\tilde{s}) \qquad (48)$$

and Property 2 holds. Property 1 holds trivially because $c_1$ is the first constructed clustering.

**Induction Hypothesis:** For a clustering $c_i$ both Property 1 and Property 2 hold.

**Induction Step:** To see why Property 1 and 2 hold for a clustering $c_{i+1}$, we first denote prediction errors with a vector $\boldsymbol{\delta}_i$ and

$$\widehat{\boldsymbol{\psi}}_i^\pi(s,a) = \boldsymbol{\psi}_i^\pi(s,a) + \boldsymbol{\delta}_i(s,a). \qquad (49)$$

If two states $s$ and $\tilde{s}$ are merged into the same partition by a maximally-compressed-reward-predictive representation (and have equal SFs $\boldsymbol{\psi}_*^\pi$), then

$$||\widehat{\boldsymbol{\psi}}_i^\pi(s,a) - \widehat{\boldsymbol{\psi}}_i^\pi(\tilde{s},a)|| \tag{50}$$

$$\leq ||\boldsymbol{\psi}_i^\pi(s,a) - \boldsymbol{\psi}_i^\pi(\tilde{s},a)|| + ||\boldsymbol{\delta}_i(s,a) - \boldsymbol{\delta}_i(\tilde{s},a)|| \qquad \text{(by substituting (49) and triangle ineq.)} \tag{51}$$

$$= ||\boldsymbol{\Phi}_i\boldsymbol{\psi}_*^\pi(s,a) - \boldsymbol{\Phi}_i\boldsymbol{\psi}_*^\pi(\tilde{s},a)|| + \underbrace{||\boldsymbol{\delta}_i(s,a) - \boldsymbol{\delta}_i(\tilde{s},a)||}_{\leq \frac{\varepsilon_\psi}{2} + \frac{\varepsilon_\psi}{2} \text{ by Assmpt. 1}} \qquad \text{(by Lemma 2)} \tag{52}$$

$$\leq ||\boldsymbol{\Phi}_i|| \cdot \underbrace{||\boldsymbol{\psi}_*^\pi(s,a) - \boldsymbol{\psi}_*^\pi(\tilde{s},a)||}_{= \,\mathbf{0} \text{ by choice of } s \text{ and } \tilde{s}} + \varepsilon_\psi \tag{53}$$

$$= \varepsilon_\psi. \tag{54}$$

Consequently,

$$c^*(s) = c^*(\tilde{s}) \implies ||\boldsymbol{f}_i(s,a) - \boldsymbol{f}_i(\tilde{s},a)|| \leq \varepsilon_\psi \implies c_{i+1}(s) = c_{i+1}(\tilde{s}). \tag{55}$$

By inversion of the implication in line (55), the Reward-predictive Splitting Property 2 holds. Furthermore, because the matching condition in line (16) holds, we have for any two states

$$c_i(s) \neq c_i(\tilde{s}) \implies ||\boldsymbol{\psi}_i^\pi(s,a) - \boldsymbol{\psi}_i^\pi(\tilde{s},a)|| > 3\varepsilon_\psi. \tag{56}$$

Consequently,

$$||\boldsymbol{f}_i(s,a) - \boldsymbol{f}_i(\tilde{s},a)|| = ||(\boldsymbol{\psi}_i^\pi(s,a) - \boldsymbol{\psi}_i^\pi(\tilde{s},a)) - (\boldsymbol{\delta}_i(\tilde{s},a) - \boldsymbol{\delta}_i(s,a))|| \tag{57}$$

$$\geq \Big| \underbrace{||\boldsymbol{\psi}_i^\pi(s,a) - \boldsymbol{\psi}_i^\pi(\tilde{s},a)||}_{>3\varepsilon_\psi} - \underbrace{||\boldsymbol{\delta}_i(\tilde{s},a) - \boldsymbol{\delta}_i(s,a)||}_{\leq 2\varepsilon_\psi} \Big| \qquad \text{(by inverse triangle ineq.)} \tag{58}$$

$$> 3\varepsilon_\psi - 2\varepsilon_\psi = \varepsilon_\psi. \tag{59}$$

Therefore, $c_{i+1}(s) \neq c_{i+1}(\tilde{s})$ and the Refinement Property 1 holds as well.

Lastly, the clustering $c_T$ returned by Algorithm 1 satisfies the conditions outlined in Definition 2. Because the Refinement Property 1 holds at every iteration, we have by line (47) that

$$\left| \boldsymbol{e}_{c_T(s)}^\top \boldsymbol{w}_a - \mathbb{E}_p[r(s,a,s')|s,a] \right| \leq \varepsilon_r \tag{60}$$

and therefore $c_T$ satisfies the bound in line (24). Furthermore, because Algorithm 1 terminates when $c_T$ and $c_{T-1}$ are identical, we have that

$$c_T(s) = c_T(\tilde{s}) \iff \left||\widehat{\boldsymbol{\psi}}_T^\pi(s,a) - \widehat{\boldsymbol{\psi}}_T^\pi(\tilde{s},a)\right|| \leq \varepsilon_\psi. \tag{61}$$

For this clustering, we can construct a set of matrices $\{\widehat{\boldsymbol{F}}_a\}_{a\in\mathcal{A}}$ by averaging the predicted SFs such that every row

$$\widehat{\boldsymbol{F}}_a(i) = \frac{1}{|\{s : c_T(s) = i\}|} \sum_{s:c_T(s)=i} \widehat{\boldsymbol{\psi}}_T^\pi(s,a). \tag{62}$$

For every observed state-action pair $(s,a)$

$$\left||\boldsymbol{e}_{c_T(s)}^\top \widehat{\boldsymbol{F}}_a - \boldsymbol{\psi}_T^\pi(s,a)\right|| = \left||\boldsymbol{e}_{c_T(s)}^\top \widehat{\boldsymbol{F}}_a - \widehat{\boldsymbol{\psi}}_i^\pi(s,a) + \boldsymbol{\delta}_i(s,a)\right|| \qquad \text{(by line (49))} \tag{63}$$

$$\leq \underbrace{\left||\boldsymbol{e}_{c_T(s)}^\top \widehat{\boldsymbol{F}}_a - \widehat{\boldsymbol{\psi}}_i^\pi(s,a)\right||}_{\leq \varepsilon_\psi \text{ by (62)}} + \underbrace{\left||\boldsymbol{\delta}_i(s,a)\right||}_{\leq \frac{\varepsilon_\psi}{2} \text{ by Assmpt 1}} \tag{64}$$

$$\leq \frac{3}{2}\varepsilon_\psi \tag{65}$$

and therefore the SF condition in line (25) holds as well (up to a rescaling of the $\varepsilon_\psi$ hyper-parameter).

$\square$

## Appendix C   Experiments

### C.1   Reward-predictive clustering experiments

In Section 4, the clustering algorithm was run on a fixed trajectory dataset that was generated by selecting actions uniformly at random. In the Column World task, a start state was sampled uniformly at random from the right column. In the Combination Lock task the start state was always the combination $(0, 0, 0)$. MNIST images were always sampled uniformly at random from the training or test sets (depending on the experiment phase).

For the Column World experiment a three layer fully connected neural network was used with ReLU activation functions. The two hidden layers have a dimension of 1000 (the output dimension depends on the number of latent states and actions). In the Combination Lock experiment the ResNet18 architecture was used by first reshaping the state image into a stack of three digit images and then feeding this image into the ResNet18 model. For all experiments the weights of the ResNet18 model were initialized at random (we did not use a pre-trained model). The 1000 dimensional output of this model was then passed through a ReLU activation function and then through a linear layer. The output dimension varied depending on the quantity the network is trained to predict during clustering. Only the top-most linear layer was re-trained between different refinement iterations, the weights of the lower layers (e.g. the ResNet18 model) were re-used across different refinement iterations. All experiments were implemented in PyTorch (Paszke et al., 2019) and all neural networks were optimized using the Adam optimizer (Kingma & Ba, 2014). We always used PyTorch's default network weight initialization heuristics and default values for the optimizer and only varied the learning rate. Mini-batches were sampled by shuffling the data set at the beginning of every epoch. Table 1 lists the used hyper-parameter.

Table 1: Hyper-parameter settings for both clustering algorithms

| Parameter | Column World | Combination Lock |
|---|---|---|
| Batch size | 32 | 256 |
| Epochs, reward refinement | 5 | 10 |
| Epochs, SF refinement | 5 | 20 |
| Epochs, representation network training | 5 | 20 |
| Learning rate | 0.005 | 0.001 |
| $\varepsilon_r$ | 0.5 | 0.4 |
| $\varepsilon_\psi$ | 1.0 | 0.8 |
| Spurious latent state filter fraction | 0.01 | 0.0025 |
| Number of training trajectories | 1000 | 10000 |

### C.2   DQN experiments

All experiments in Figure 6 were repeated 20 times and each agent spent 100 episodes in each task. To select actions, an $\varepsilon$-greedy exploration strategy was used that selects actions with $\varepsilon$ probability greedily (with respect to the Q-value predictions) and with $1 - \varepsilon$ actions are selected uniformly at random. During the first episode in each training and test task, $\varepsilon = 0$ and the $\varepsilon$ was linearly increase to 1 within 10 time steps. The DQN agent always used a Q-network architecture consisting of the ResNet18 architecture (with random weight initialization), a ReLU activation function, and then a fully connected layer to predict Q-values for each action (as illustrated in Figure 6(b)). Table 2 outlines the hyper-parameters that were fine tuned for the combination lock training task. These hyper-parameters were then re-used for all DQN variants used in Section 4.2.

Table 2: Hyper-parameter sweep results for DQN on the combination lock training task.

| Parameter | Tested Values | Best Setting (highest reward-per-step score) |
|---|---|---|
| Learning rate | $10^{-4}$, $10^{-3}$, $10^{-2}$, $10^{-1}$ | $10^{-3}$ |
| Batch size | 100, 200, 500 | 200 |
| Buffer size | 100, 1000, 10000 | 10000 |
| Exploration episodes | 5, 10, 20, 50, 80 | 10 |

