# OpenReview forum: "Reward-Predictive Clustering"
_TMLR — Rejected by TMLR_

### Review · Reviewer_kkWR · 2022-12-05

**Summary Of Contributions:**

This paper builds off of the work on reward-predictive representation learning (Lehnert et al. 2020; Lenhert & Littman, 2020), extending these ideas to the deep learning setting. This is done by introducing a clustering algorithm that clusters together observations that lead to the same sequence of rewards. This is done by using successor features (SFs) to represent states before they are clustered, by defining a cross-entropy loss that makes the problem closer to a classification problem than a regression problem, and so on. The clustering algorithm automatically identifies the number of clusters needed.

Aside from some theoretical guarantees based on strong assumptions about the clustering algorithm being quasi-perfect, the paper presents results in simulation in combination-lock problems that use MNIST digits to represent the numbers in the dials.


**Audience:**

Yes

**Broader Impact Concerns:**

I don't have any concerns.

**Claims And Evidence:**

Yes

**Requested Changes:**

I have two question I want the authors to answer and to make it clear in the paper:

- How much does the proposed assumption relies/depends on the environment being quasi-determinsitic?
- In lines 394-400 it is said how the proposed approach re-uses the pre-trained representation network, not performing weight updates on the representation network. Was this approach tried in the other baselines as well? It seems to me the answer is no, but I wonder if not updating the representation network wouldn’t make it altogether better.

Aside from these two questions (and the necessary actions depending on the answers), I would like to see the performance of the different baselines I mentioned for Figure 6(c), specifically: a bisimulation-based method, a meta-learning method, and SF-based GPI, with a discussion about their failing patterns. Finally, I would like to see an analysis of how performance of reward-predictive methods degrades as the quality of the clustering decays.


**Strengths And Weaknesses:**

The paper tackles an important problem in the field. State abstraction is a promising way to lead to faster credit assignment and knowledge reuse. The paper is sound and it clearly delineates the problem it is tackling and the assumptions it is making.

Overall, I do really worry about the assumptions being made in terms of the accuracy of the clustering algorithm. I find the justification based on the double descent weak as that does not guarantee perfect accuracy. In fact, in reinforcement learning problems, it seems to me that the assumption that clustering algorithm can be near-perfect is based on the assumption that the agent has enough capacity to model the whole world. This is quite problematic, in most problems (certainly those in the real world), the environment is much bigger than the agent and the agent needs to accept the fact it cannot model everything. I do understand that oftentimes we need to depart away from theoretical results in order to deploy the methods we design, and I wish that had been evaluated. Specifically, I wish there was an analysis on the performance of the proposed method when clustering is inaccurate at varying degrees. How gracefully does performance decay?

I fight against the urge to ask for comparisons in different domains, more standard ones, such as those used in the SFs paper, and I don’t condition the acceptance of the paper on this, but I do think the paper would benefit from more thorough empirical validation. Importantly, and I find this crucial, I feel there are important baselines missing in the paper. The paper does discuss (at the end) methods based on bisimulation metrics, for example, but it never uses any of those methods as baseline as much as they are so similar from using the SFs + reward predictions to cluster states together. I believe a good baseline here would be something like MICo. Also, I was very confused by the discussion around Figure 6(c). MAML and SF-based GPI were rightly discussed in the main text but from what I can tell they were not used as baselines. Why is that?

Finally, the text can be improved in some parts.
- Line 96: The SR can be defined wrt to any policy, it is not defined only in terms of the optimal policy.
- Line 159: Is $L_i$ a random variable or are the trajectories assumed to have fixed length?
- Line 170: “rewards are typically sparse in an RL task”. The reinforcement learning problem formulation defines a reward that is observed at each time step, rewards are never sparse. Rewards might be zero most of the time with a +1 upon completion, but is this different from a reward -1 at each time step with a 0 upon completion? Also, some of the most prominent benchmarks in reinforcement learning (e.g., MuJoCo) have non-zero rewards at almost every time step.
- Caption of Figure 2 and lines 177-189: How can we ever ensure that generalization is going to happen as described? It seems wishful thinking.
- Equation 9: I believe the hat was not defined in the main paper. What does the hat stand for on top of the $\Psi$?
- Algorithm 1:  I believe the function $H(\cdot, \cdot)$ was not defined in the main paper.
- Line 277: In English, didactic can have a bad connotation, was this the intended use of the word?



References:
Castro et al.: MICo: Improved representations via sampling-based state similarity for Markov decision processes. NeurIPS 2021.

---

### Review · Reviewer_FFhJ · 2023-01-13

**Summary Of Contributions:**

The submission investigates reward predictive state abstractions in a deep learning setting. Its first contribution is a clustering algorithm enabling to port the reward predictive state abstraction to a deep learning setting. Its second contribution is a theorem proving the convergence of the algorithm under a set of assumptions. Finally, a broad set of experiments empirically validate the efficiency of the approach.

**Audience:**

Yes

**Broader Impact Concerns:**

no concern

**Claims And Evidence:**

No

**Requested Changes:**

Please address the major issues enumerated above.

**Strengths And Weaknesses:**

Adapting the reward predictive state abstraction to a deep learning setting is interesting, promising, and novel as far as I can tell, and the submission is generally well exposed. Nevertheless, it presents several major issues:
* The motivation could be improved.
   * L21-28: I am not convinced that reward predictive state abstraction can be claimed as advantageous for reusability to other tasks, since the reward prediction is tied to the task.
   * L54-55: It is unclear 1/ what is a compressed latent space and 2/ why we would need this
   * L63-70: this task seems designed for the reward predictive bias to be beneficial. We could imagine the same environment where the action moving to the right depends on the row number and then the clustering of states would be ill advised or am I missing something?
* Some claims are not sufficiently supported, and they may appear overstated:
   * abstract: "maximally compresses" => it is not maximal.
   * L127-128: what do you mean exactly by that? Why? Doesn't non-linear clustering induce a representation function that is non-linear?
   * L170-173: strange choices and claims. It rather leaves me the impression that the real reason for this choice is that the authors could not make it work otherwise.
   * Eq9: These clustering objectives do not seem to be adapted to stochasticity, either in rewards, transitions or behavioral policy.
   * Thm1: The assumption defined in eq16 should be made more visible in introduction. This is drastic and a bit hidden in the current exposition (even though, the requirement is clear in the theorem).

Minor comments/typos:
* L42: Besides => besides
* L42: decide between => choose among
* L61: the reward function looks deterministic. You should make it clear.
* Fig1: given otherwise => reached otherwise
* L126-156: too long of a paragraph, I would break it down.
* Eq4 and Eq8: shouldn't the difference be squared?
* Alg1: appears too late.
* Eq16: γ<1/2 is implied by the second inequality.

---

### Review · Reviewer_LF5r · 2023-01-29

**Summary Of Contributions:**

The paper proposes an approach for learning representations in RL that are indicative of the future reward. They propose an iterative  refinement approach for learning this space by learning successor features, and using them to cluster states. The authors include a proof that the resulting representation maximally compresses the observations, and include some experiments for OOD generalization.

**Audience:**

Yes

**Claims And Evidence:**

No

**Requested Changes:**

Please address the concerns listed in the weaknesses section. Specifically, please evaluate the approach on distract envs from mujoco [1,2], and even better if you can show better efficiency/transfer results with this representation on more realistic simulation envs [3-4]. Also please add comparisons to DBC[1] and other prior work listed above. DBC is specially pertinent since it addresses the same question - how to build representations that are informative of future reward.




[1] - Learning invariant representations for reinforcement learning without reconstruction
[2] - The Distracting Control Suite – A Challenging Benchmark for Reinforcement Learning from Pixels
[3] - Meta-World: A Benchmark and Evaluation for Multi-Task and Meta Reinforcement Learning
[4] - robosuite: A Modular Simulation Framework and Benchmark for Robot Learning


**Strengths And Weaknesses:**

Strengths -
The paper considers an important problem - that of learning effective representations for RL. Learning control policies from high dimensional observations like images is quite sample inefficient and useful representations will help learn faster and also transfer/generalize.


Weaknesses -
1. The paper does not compare to prior work that seeks the tackle the same problem - representation learning in RL to predict future reward. Zhang et al. [1] learn a bisimulation metric representation (DBC), where states that get similar expected return are embedded to the same representations. Further, this work is tested on a wide suite of visual RL environments, including visual robots in Mujoco requiring continuous control and self-driving settings in a carla simulator, both which are much more complex than the settings considered in this paper. Further the proposed paper also does not compare to other works in representation learning for visual RL, like [2,3,4]. The authors only compare against DQN on a relatively simplistic combination lock environment that they propose. In order to better understand the significance and benefit of the proposed approach it needs to be evaluated a lot better.

2. Having a discrete number of clusters seems like an important limitation, especially since DBC[1] learns a continuous embedding of the observation, and has been shown to work quite well for RL tasks.


[1] : Learning invariant representations for reinforcement learning without reconstruction
[2] :  DeepMDP: Learning continuous latent space models for representation learning.
[3] : Curl: Contrastive unsupervised representations for reinforcement learning
[4] : Image Augmentation Is All You Need: Regularizing Deep Reinforcement Learning from Pixels

---

### Comment · Reviewer_kkWR · 2023-02-16
**New version of the paper**

Dear authors:

Several of my concerns about the paper have been eased by your responses. However, I was expecting to see the revised version of the paper submitted alongside the comments, otherwise I can't comment/evaluate the revision itself.

About MICo, I am very looking forward to seeing those results. The main reason I asked about them is not because they approach the same problem in the same way, but exactly because they approach the learning problem differently. This comparison allows us to get empirical evidence for which approach might be better when tackling such a problem.

Finally, I apologize for my own typo, I believe the function H in Algorithm 1 was not defined.

---

### Decision · Action_Editors · 2023-03-14

**Recommendation:** Reject

**Comment:**

The lack of empirical support for the claims was noted by all of the reviewers, who consistently asked for better baselines, more diverse tasks, and more clarity on limitations. The authors responded to the reviewer comments, but have said that fully addressing them would require more time. (They also dismissed some of the reviewer concerns, which is not ideal). Given these issues, all three reviewers recommended rejection. As such, the AE was led to a rejection decision. However, if the authors can address the reviewers' concerns more fully in the future, they are welcome to resubmit the paper at a later time. If they do so, note that it will likely be sent to the same three reviewers.

**Audience:**

This paper would be interesting to the TMLR audience, as it provides a proposed mechanism for representation learning in deep RL agents that could improve learning speed and generalization.

**Claims And Evidence:**

This paper presents a novel algorithm for training deep neural networks to cluster inputs based on reward prediction. The goal is to develop better means for deep RL agents to learn abstractions that can accelerate learning in new contexts. The central claim is that their algorithm can speed up learning in high-dimensional control tasks by helping deep RL agents to exhibit improved out-of-distribution transfer.

The authors provide analytical results showing that their proposed clustering algorithm will converge to compressed reward predictive representations, and some empirical results on a combination lock task with comparisons to a DQN agent. However, they do not provide convincing evidence that their algorithm speeds up transfer learning beyond other appropriate baseline representation learning approaches, particularly bisimulation approaches. As well, they do not test the algorithm on tasks that are not designed by themselves, and which may be particularly well-suited for reward predictive representations. Thus, the claims are not fully supported by clear, convincing evidence.